# No Press Diplomacy: Modeling Multi-Agent Gameplay

**Philip Paquette** [1]
pcpaquette@gmail.com

**Yuchen Lu** [1]
luyuchen.paul@gmail.com

**Steven Bocco** [1]
stevenbocco@gmail.com

**Max O. Smith** [3]
max.olan.smith@gmail.com

**Satya Ortiz-Gagné** [1]
s.ortizgagne@gmail.com

**Jonathan K. Kummerfeld** [3]
jkummerf@umich.edu

**Satinder Singh** [3]
baveja@umich.edu

**Joelle Pineau** [2]
jpineau@cs.mcgill.ca

**Aaron Courville** [1]
aaron.courville@gmail.com

## Abstract

Diplomacy is a seven-player non-stochastic, non-cooperative game, where agents acquire resources through a mix of teamwork and betrayal. Reliance on trust and coordination makes Diplomacy the first non-cooperative multi-agent benchmark for complex sequential social dilemmas in a rich environment. In this work, we focus on training an agent that learns to play the No Press version of Diplomacy where there is no dedicated communication channel between players. We present *DipNet*, a neural-network-based policy model for No Press Diplomacy. The model was trained on a new dataset of more than 150,000 human games. Our model is trained by supervised learning (SL) from expert trajectories, which is then used to initialize a reinforcement learning (RL) agent trained through self-play. Both the SL and RL agents demonstrate state-of-the-art No Press performance by beating popular rule-based bots.

## 1 Introduction

Diplomacy is a seven-player game where players attempt to acquire a majority of supply centers across Europe. To acquire supply centers, players can coordinate their units with other players through dialogue or signaling. Coordination can be risky, because players can lie and even betray each other. Reliance on trust and negotiation makes Diplomacy the first non-cooperative multi-agent benchmark for complex sequential social dilemmas in a rich environment.

Sequential social dilemmas (SSD) are situations where one individual experiences conflict between self- and collective-interest over repeated interactions [1]. In Diplomacy, players are faced with a SSD in each phase of the game. Should I help another player? Do I betray them? Will I need their help later? The actions they choose will be visible to the other players and influence how other players interact with them later in the game. The outcomes of each interaction are non-stochastic. This characteristic sets Diplomacy apart from previous benchmarks where players could additionally

---

1 Mila, University of Montreal
2 Mila, McGill University
3 University of Michigan
§ Dataset and code can be found at https://github.com/diplomacy/research

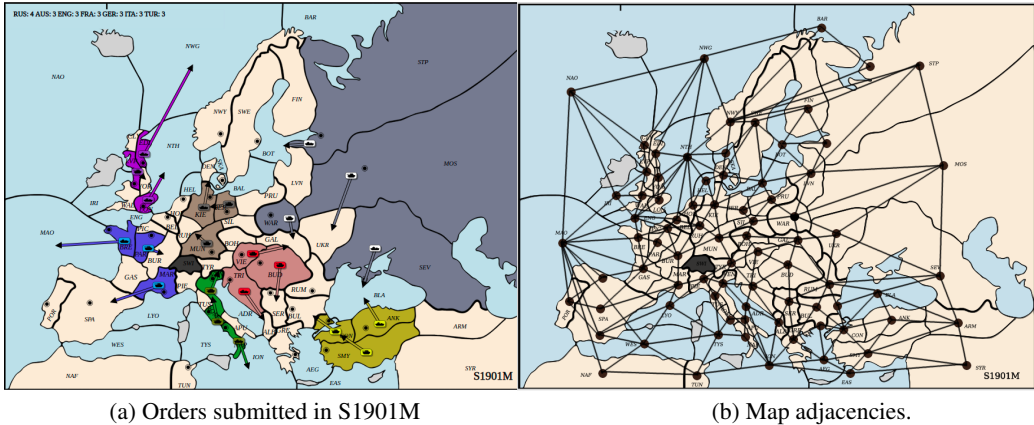

(a) Orders submitted in S1901M                    (b) Map adjacencies.

Figure 1: The standard Diplomacy map

rely on chance to win [2, 3, 4, 5, 6]. Instead, players must put their faith in other players and not in the game's mechanics (e.g. having a player role a critical hit).

Diplomacy is also one of the first SSD games to feature a rich environment. A single player may have up to 34 units, with each unit having an average of 26 possible actions. This astronomical action space makes planning and search intractable. Despite this, thinking at multiple time scales is an important aspect of Diplomacy. Agents need to be able to form a high-level long-term strategy (e.g. with whom to form alliances) and have a very short-term execution plan for their strategy (e.g. what units should I move in the next turn). Agents must also be able to adapt their plans, and beliefs about others (e.g. trustworthiness) depending on how the game unfolds.

In this work, we focus on training an agent that learns to play the No Press version of Diplomacy. The No Press version does not allow agents to communicate with each other using an explicit communication channel. Communication between agents still occurs through signalling in actions [7, 2]. This allows us to first focus on the key problem of having an agent that has learned the game mechanics, without introducing the additional complexity of learning natural language and learning complex interactions between agents.

We present *DipNet*, a fully end-to-end trained neural-network-based policy model for No Press Diplomacy. To train our architecture, we collect the first large scale dataset of Diplomacy, containing more than 150,000 games. We also develop a game engine that is compatible with DAIDE [8], a research framework developed by the Diplomacy research community, and that enables us to compare with previous rule-based state-of-the-art bots from the community [9]. Our agent is trained with supervised learning over the expert trajectories. Its parameters are then used to initialize a reinforcement learning agent trained through self-play.

In order to better evaluate the performance of agents, we run a tournament among different variants of the model as well as baselines, and compute the TrueSkill score [10]. Our tournament shows that both our supervised learning (SL) and reinforcement learning (RL) agents consistently beat baseline rule-based agents. In order to further demonstrate the affect of architecture design, we perform an ablation study with different variants of the model, and find that our architecture has higher prediction accuracy for support orders even in longer sequences. This ability suggests that our model is able to achieve tactical coordination with multiple units. Finally we perform a coalition analysis by computing the ratio of cross-power support, which is one of the main methods for players to cooperate with each other. Our results suggest that our architecture is able to issue more effective cross-power orders.

## 2   No Press Diplomacy: Game Overview

Diplomacy is a game where seven European powers (Austria, England, France, Germany, Italy, Russia, and Turkey) are competing over supply centers in Europe at the beginning of the 20th century.

There are 34 supply centers in the game scattered across 75 provinces (board positions, including water). A power interacts with the game by issuing orders to army and fleet units. The game is split into years (starting in 1901) and each year has 5 phases: Spring Movement, Spring Retreat, Fall Movement, Fall Retreat, and Winter Adjustment.

**Movements.** There are 4 possible orders during a movement phase: Hold, Move, Support, and Convoy. A hold order is used by a unit to defend the province it is occupying. Hold is the default order for a unit if no orders are submitted. A move order is used by a unit to attack an adjacent province. Armies can move to any adjacent land or coastal province, while fleets can move to water or coastal provinces by following a coast.

Support orders can be given by any power to increase the attack strength of a moving unit or to increase the defensive strength of a unit holding, supporting, or convoying. Supporting a moving unit is only possible if the unit issuing the support order can reach the destination of the supported move (e.g. Marseille can support Paris moving to Burgundy, because an army in Marseille could move to Burgundy). If the supporting unit is attacked, its support is unsuccessful.

It is possible for an army unit to move over several water locations in one phase and attack another province by being convoyed by several fleets. A matching convoy order by the convoying fleets and a valid path of non-dislodged fleets (explained below) is required for the convoy to be successful.

**Retreats.** If an attack is successful and there is a unit in the conquered province, the unit is dislodged and is given a chance to retreat. There are 2 possible orders during a retreat phase: Retreat and Disband. A retreat order is the equivalent of a move order, but only happens during the retreat phase. A unit can only retreat to a location that is 1) unoccupied, 2) adjacent, and 3) not a standoff location (i.e. left vacant because of a failed attack). A disband order indicates that the unit at the specified province should be removed from the board. A dislodged unit is automatically disbanded if either there are no possible retreat locations, it fails to submit a retreat order during the retreat phase, or two units retreat to the same location.

**Adjustments.** The adjustment phase happens once every year. During that phase, supply centers change ownership if a unit from one power occupies a province with a supply center owned by another power. There are three possible orders during an adjustment phase: Build, Disband, and Waive. If a power has more units than supply centers, it needs to disband units. If a power has more supply centers than units, it can build additional units to match its number of supply centers. Units can only be built in a power's original supply centers (e.g. Berlin, Kiel, and Munich for Germany), and the power must still control the chosen province and it must be unoccupied. A power can also decide to waive builds, leaving them with fewer units than supply centers.

**Communication in a No Press game** In a No Press game, even if there are no messages, players can communicate between one another by using orders as signals [7]. For example, a player can declare war by positioning their units in an offensive manner, they can suggest possible moves with support and convoy orders, propose alliances with support orders, propose a draw by convoying units to Switzerland, and so on. Sometimes even invalid orders can be used as communication, e.g., Russia could order their army in St. Petersburg to support England's army in Paris moving to London. This invalid order could communicate that France should attack England, even though Paris and St. Petersburg are not adjacent to London.

**Variants** There are three important variants of the game: Press, Public Press, and No Press. In a Press game, players are allowed to communicate with one another privately. In a Public Press game, all messages are public announcements and can be seen by all players. In a No Press game, players are not allowed to send any messages. In all variants, orders are written privately and become public simultaneously, after adjudication. There are more than 100 maps available to play the game (ranging from 2 to 17 players), though the original Europe map is the most played, and as a result is the focus of this work. The final important variation is *check*, where invalid orders can be submitted (but are then not applied), versus *no-check* where only valid orders are submitted. This distinction is important, because it determines the inclusion of a side-channel for communication through invalid orders.

**Game end.** The game ends when a power is able to reach a majority of the supply centers (18/34 on the standard map), or when players agree to a draw. When a power is in the lead, it is fairly common for other players to collaborate to prevent the leading player from making further progress and to force a draw.

**Scoring system.** Points in a diplomacy game are usually computed either with 1) a draw-based scoring system (points in a draw are shared equally among all survivors), or 2) a supply-center count scoring system (points in a draw are proportional to the number of supply centers). Players in a tournament are usually ranked with a modified Elo or TrueSkill system [10][11][12][13].

## 3   Previous Work

In recent years, there has been a definite trend toward the use of games of increasingly complexity as benchmarks for AI research including: Atari [14], Go [15][16], Capture the Flag [17], Poker [3][4], Starcraft [6], and DOTA [5]. However, most of these games do not focus on communication. The benchmark most similar to our No Press Diplomacy setting is Hanabi [2], a card game that involves both communication and action. However Hanabi is fully cooperative, whereas in Diplomacy, ad hoc coalitions form and degenerate dynamically throughout the evolution of the game. We believe this makes Diplomacy unique and deserving of special attention.

Previous work on Diplomacy has focused on building rule-based agents with substantial feature engineering. DipBlue [18] is a rule-based agent that can negotiate and reason about trust. It was developed for the DipGame platform [19], a DAIDE-compatible framework [8] that also introduced a language hierarchy. DBrane [20] is a search-based bot that uses branch-and-bound search, with state evaluation to truncate as appropriate. Another work, most similar to ours, uses self-play to learn a game strategy leveraging patterns of board states [21]. Our work is the first attempt to use a data-driven method on a large-scale dataset.

Our work is also related to the learning-to-cooperate literature. In classical game theory, the Iterated Prisoner's Dilemma (IPD) has been the main focus for SSD, and a tit-for-tat strategy has been shown to be a highly effective strategy [22]. Recent work [23] has proposed an algorithm that takes into account the impact of one agent's policy on the update of the other agents. The resulting algorithm was able to achieve reciprocity and cooperation in both IPD and a more complex coin game with deep neural networks. There is also a line of work on solving social dilemmas with deep RL, which has shown that enhanced cooperation and meaningful communication can be promoted via causal inference [24], inequity aversions [25], and understanding consequences of intention [26]. However, most of this work has only been applied to simple settings. It is still an open question whether these methods could scale up to a complex domain like Diplomacy.

Our work is also related to behavioral game theory, which extends game theory to account for human cognitive biases and limitations [27]. Such behavior is observed in Diplomacy when players make non-optimal moves due to ill-conceived betrayals or personal vengeance against a perceived slight.

## 4   *DipNet*: A Generative Model of Unit Orders

### 4.1   Input Representation

Our model takes two inputs: current board state and previous phase orders. To represent the board state, we encode for each province: the type of province, whether there is a unit on that province, which power owns the unit, whether a unit can be built or removed in that province, the dislodged unit type and power, and who owns the supply center, if the province has one. If a fleet is on a coast (e.g. on the North Coast of Spain), we also record the unit information in the coast's parent province.

Previous orders are encoded in a way that helps infer which powers are allies and enemies. For instance, for the order 'A MAR S A PAR - BUR' (*Army in Marseille supports army in Paris moves to Burgundy*), we would encode: 1) 'Army' as the unit type, 2) the power owning 'A MAR', 3) 'support' as the order type, 4) the power owning 'A PAR' (i.e. the friendly power), 5) the power, if any, having either a unit on BUR or owning the BUR supply center (i.e. the opponent power), 6) the owner of the BUR supply center, if it exists. Based on our empirical findings, orders from the last movement

phase are enough to infer the current relationship between the powers. Our representation scheme is shown in Figure 2, with one vector per province.

## 4.2 Graph Convolution Network with FiLM

To take advantage of the adjacency information on the Diplomacy map, we propose to use a graph convolution-based encoder [28]. Suppose $x_{bo}^l \in \mathbb{R}^{81 \times d_{bo}^l}$ is the board state embedding produced by layer $l$ and $x_{po}^l \in \mathbb{R}^{81 \times d_{po}^l}$ is the corresponding embedding of previous orders, where $x_{bo}^0, x_{po}^0$ are the input representations described in Section 4.1. We will now describe the process for encoding the board state; the process for the previous order embedding is the same. Suppose $A$ is the normalized map adjacency matrix of $81 \times 81$. We first aggregate neighbor information by:

$$y_{bo}^l = BatchNorm(Ax_{bo}^l W_{bo} + b_{bo})$$

where $W_{bo} \in \mathbb{R}^{d_{bo}^l \times d_{bo}^{l+1}}$, $b_{bo} \in \mathbb{R}^{d_{bo}^{l+1}}$, $y_{bo}^l \in \mathbb{R}^{81 \times d_{bo}^{l+1}}$ and $BatchNorm$ is operated on the last dimension. We perform conditional batch normalization using FiLM [29, 30], which has been shown to be an effective method of fusing multimodal information in many domains [31]. Batch normalization is conditioned on the player's power $p$ and the current season $s$ (Spring, Fall, Winter).

$$\gamma_{bo}, \beta_{bo} = f_{bo}^l([p;s]) \qquad z_{bo}^l = y_{bo}^l \odot \gamma_{bo} + \beta_{bo} \tag{1}$$

where $f_l$ is a linear transformation, $\gamma, \beta \in R^{d^{l+1}}$, and both addition and multiplication are broadcast across provinces. Finally we add a $ReLU$ and residual connections [32] where possible:

$$x_{bo}^{l+1} = \begin{cases} ReLU(z_{bo}^l) + x_{bo} & d^l = d_{bo}^{l+1} \\ ReLU(z_{bo}^l) & d_{bo}^l \neq d_{bo}^{l+1} \end{cases}$$

The board state and the previous orders are both encoded through $L$ of these blocks, and there is no weight sharing. Concatenation is performed at the end, giving $h_{enc} = [x_{bo}^L, x_{po}^L]$ where $h_{enc}^i$ is the final embedding of the province with index $i$. We choose $L = 16$ in our experiment.

## 4.3 Decoder

In order to achieve coordination between units, sequential decoding is required. However there is no natural sequential ordering. We hypothesize that orders are usually given to a cluster of nearby units, and therefore processing neighbouring units together would be effective. We used a top-left to bottom-right ordering based on topological sorting, aiming to prevent jumping across the map during decoding.

Suppose $i^t$ is the index of the province requiring an order at time $t$, we use an LSTM to decode its order $o^t$ by

$$h_{dec}^t = \text{LSTM}(h_{dec}^{t-1}, [h_{enc}^{i^t}; o^{t-1}]) \tag{2}$$

Then we apply a mask to only get valid possible orders for that location on the current board:

$$o^t = \text{MaskedSoftmax}(h_{dec}^t) \tag{3}$$

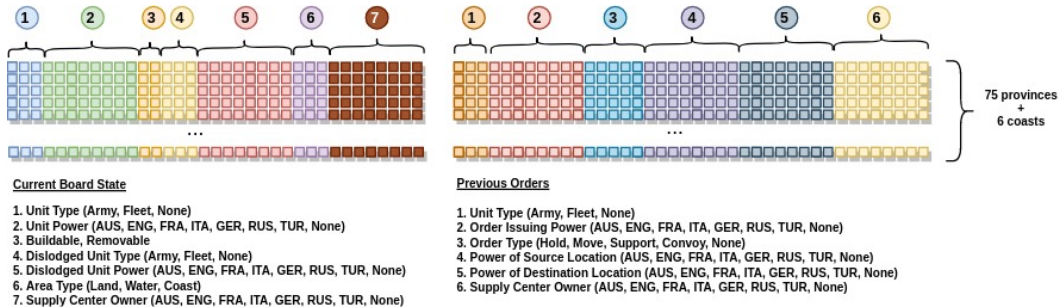

Figure 2: Encoding of the board state and previous orders.

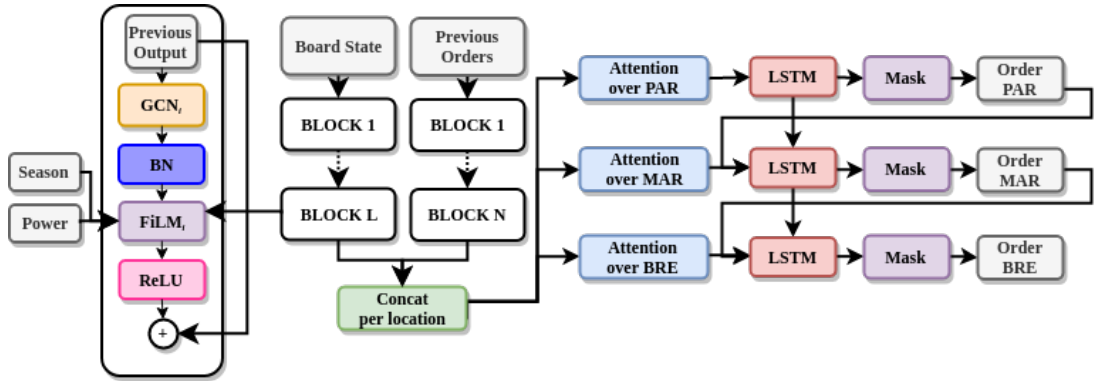

Figure 3: DipNet architecture

# 5 Datasets and Game Engine

Our dataset is generated by aggregating 156,468 anonymized human games. We also develop an open source game engine for this dataset to standardize its format and rule out invalid orders. The dataset contains 33,279 No Press games, 1,290 Public Press games, 105,266 Press games (messages are not included), and 16,633 games not played on the standard map. We are going to release the dataset along with the game engine[5]. Detailed dataset statistics are shown in Table 1.

The game engine is also integrated with the Diplomacy Artificial Intelligence Development Environment (DAIDE) [8], an AI framework from the Diplomacy community. This enables us to compare with several state-of-the-art rule-based bots [9, 18] that have been developed on DAIDE. DAIDE also has a progression of 14 symbolic language levels (from 0 to 130) for negotiation and communication, which could be potentially useful for research on Press Diplomacy. Each level defines what tokens are allowed to be exchanged by agents. For instance, a No Press bot would be considered level 0, while a level 20 bot can propose peace, alliances, and orders.

# 6 Experiments

## 6.1 Supervised Learning

We first present our supervised learning results. Our test set is composed of the last 5% of games sorted by game id in alphabetical order. To measure the impact of each model component, we ran an ablation study. The results are presented in Table 2. We evaluate the model with both greedy decoding and teacher forcing. We measure the accuracy of each unit-order (e.g. 'A PAR - BUR'), and the accuracy of the complete set of orders for a power (e.g. 'A PAR - BUR', 'F BRE - MAO'). We

Table 1: Dataset statistics

| | | | | Survival rate for opponents | | | | | | |
|---|---|---|---|---|---|---|---|---|---|---|
| | Win% | Draw% | Defeated% | AUS | ENG | FRA | GER | ITA | RUS | TUR |
| Austria | 4.3% | 33.4% | 48.1% | 100% | 79% | 62% | 55% | 40% | 29% | 15% |
| England | 4.6% | 43.7% | 29.1% | 47% | 100% | 30% | 16% | 49% | 33% | 80% |
| France | 6.1% | 43.8% | 25.7% | 40% | 26% | 100% | 22% | 45% | 59% | 77% |
| Germany | 5.3% | 35.9% | 40.4% | 44% | 26% | 39% | 100% | 61% | 27% | 80% |
| Italy | 3.6% | 36.5% | 40.2% | 15% | 65% | 56% | 61% | 100% | 56% | 25% |
| Russia | 6.6% | 35.2% | 39.8% | 25% | 52% | 77% | 38% | 63% | 100% | 42% |
| Turkey | 7.2% | 43.1% | 26.0% | 9% | 78% | 71% | 56% | 23% | 31% | 100% |
| **Total** | 39.9% | 60.1% | | 37% | 59% | 65% | 49% | 51% | 50% | 64% |

Table 2: Evaluation of supervised models: Predicting human orders.

| Model | Accuracy per unit-order | | Accuracy for all orders | |
|---|---|---|---|---|
| | Teacher forcing | Greedy | Teacher forcing | Greedy |
| DipNet | **61.3%** | **47.5%** | **23.5%** | **23.5%** |
| Untrained | 6.6% | 6.4% | 4.2% | 4.2% |
| Without FiLM | 60.7% | 47.0% | 22.9% | 22.9% |
| Masked Decoder (No Board) | 47.8% | 26.5% | 14.7% | 14.7% |
| Board State Only | 60.3% | 45.6% | 22.9% | 23.0% |
| Average Embedding | 59.9% | 46.2% | 23.2% | 23.2% |

find that our untrained model with a masked decoder performs better than the random model, which suggests the effectiveness of masking out invalid orders. We observe a small drop in performance when we only provide the board state. We also observe a performance drop when we use the average embedding over all locations as input to the LSTM decoder (rather than using attention based on the location the current order is being generated for).

To further demonstrate the difference between these variants we focus on the model's ability to predict support orders, which are a crucial element for successful unit coordination. Table 3 shows accuracy on this order type, separated based on the position of the unit in the prediction sequence. We can see that although the performance of different variants of the model are close to each other when predicting support for the first unit, the difference is larger when predicting support for the 16$^{\text{th}}$ unit. This indicates that our architecture helps DipNet maintain tactical coordination across multiple units.

Table 3: Comparison of the models' ability to predict support orders with greedy decoding.

| | Support Accuracy | |
|---|---|---|
| | 1$^{\text{st}}$ location | 16$^{\text{th}}$ location |
| DipNet | **40.3%** | **32.2%** |
| Board State Only | 38.5% | 25.9% |
| Without FiLM | 40.0% | 30.3% |
| Average Embedding | 39.1% | 27.9% |

## 6.2 Reinforcement Learning and Self-play

We train *DipNet* with self-play (same model for all powers, with shared updates) using an A2C architecture [14] with n-step (n=15) returns for approximately 20,000 updates (approx. 1 million steps). As a reward function, we use the average of (1) a local reward function (+1/-1 when a supply center is gained or lost (updated every phase and not just in Winter)), and (2) a terminal reward function (for a solo victory, the winner gets 34 points; for a draw, the 34 points are divided

Table 4: Diplomacy agents comparison when played against each other, with one agent controlling one power and the other six powers controlled by copies of the other agent.

| Agent A (1x) | Agent B (6x) | TrueSkill A-B | % Win | % Most SC | % Survived | % Defeated | # Games |
|---|---|---|---|---|---|---|---|
| SL DipNet | Random | 28.1 - 19.7 | 100.0% | 0.0% | 0.0% | 0.0% | 1,000 |
| SL DipNet | GreedyBot | 28.1 - 20.9 | 97.8% | 1.2% | 1.0% | 0.0% | 1,000 |
| SL DipNet | Dumbbot | 28.1 - 19.2 | 74.8% | 9.2% | 15.4% | 0.6% | 950 |
| SL DipNet | Albert 6.0 | 28.1 - 24.5 | 28.9% | 5.3% | 42.8% | 23.1% | 208 |
| SL DipNet | RL DipNet | 28.1 - 27.4 | 6.2% | 0.3% | 41.4% | 52.1% | 1,000 |
| Random | SL DipNet | 19.7 - 28.1 | 0.0% | 0.0% | 4.4% | 95.6% | 1,000 |
| GreedyBot | SL DipNet | 20.9 - 28.1 | 0.0% | 0.0% | 8.5% | 91.5% | 1,000 |
| Dumbbot | SL DipNet | 19.2 - 28.1 | 0.0% | 0.1% | 5.0% | 95.0% | 950 |
| Albert 6.0 | SL DipNet | 24.5 - 28.1 | 5.8% | 0.4% | 12.6% | 81.3% | 278 |
| RL DipNet | SL DipNet | 27.4 - 28.1 | 14.0% | 3.5% | 42.9% | 39.6% | 1,000 |

proportionally to the number of supply centers). The policy is pre-trained using DipNet SL described above. We also used a value function pre-trained on human games by predicting the final rewards.

The opponents we have used to evaluate our agents were: **(1) Random.** This agent selects an action per unit uniformly at random from the list of valid orders. **(2) GreedyBot.** This agent greedily tries to conquer neighbouring supply centers and is not able to support any attacks. **(3) Dumbbot [33].** This rule-based bot computes a value for each province, ranks orders using computed province values and uses rules to maintain coordination. **(4) Albert Level 0 [9].** Albert is the current state-of-the-art agent. It evaluates the probability of success of orders, and builds alliances and trust between powers, even without messages. To evaluate performance, we run a 1-vs-6 tournament where each game is structured with one power controlled by one agent and the other six controlled by copies of another agent. We also run another tournament where each player is randomly sampled from our model pools and compute TrueSkill scores for these models [10]. We report both the 1-vs-6 results and the TrueSkill scores in Table 4. From the TrueSkill score we can see both the SL (28.1) and RL (27.4) versions of DipNet consistently beat the baseline models as well as Albert (24.5), the previous state-of-art bot. Although there is no significant difference in TrueSkill between SL and RL, the performance of RL vs 6 SL is better than SL vs 6 RL with an increasing win rate.

### 6.3   Coalition Analysis

In the No Press games, cross-power support is the major method for players to signal and coordinate with each other for mutual benefit. In light of this, we propose a coalition analysis method to further understand agents' behavior. We define a cross-power support (X-support) as being when a power supports a foreign power, and we define an effective cross-power support as being a cross-power order support without which the supported attack or defense would fail:

$$X\text{-}support\text{-}ratio = \frac{\#X\text{-}support}{\#support}, \qquad Eff\text{-}X\text{-}support\text{-}ratio = \frac{\#Effective\ X\text{-}support}{\#X\text{-}support}$$

The *X-support-ratio* reflects how frequently the support order is used for cooperation/communication, while the *Eff-X-support-ratio* reflects the efficiency or utility of cooperation. We launch 1000 games with our model variants for all powers and compute this ratio for each one. Our results are shown in Table 5.

For human games, across different game variants, there is only minor variations in the *X-support-ratio*, but the *Eff-X-support-ratio* varies substantially. This shows that when people are allowed to communicate, their effectiveness in cooperation increases, which is consistent with previous results that cheap talk promotes cooperation for agents with aligned interests [34, 35]. In terms of agent variants, although RL and SL models show similar TrueSkill scores, their behavior is very different. RL agents seem to be less effective at cooperation but have more frequent cross-power support. This decrease in effective cooperation is also consistent with past observations that naive policy gradient methods fail to learn cooperative strategies in a non-cooperative setting such as the iterated prisoner dilemma [36]. Ablations of the SL model have a similar *X-support-ratio*, but suffer from a loss in *Eff-X-support-ratio*. This further suggests that our DipNet architecture can help agents cooperate more effectively. The Masked Decoder has a very high *X-support-ratio*, suggesting that the marginal distribution of support is highest among agent games, however, it suffers from an inability to effectively cooperate (i.e. very small *Eff-X-support-ratio*). This is also expected since the Masked Decoder has no board information to understand the effect of supports.

## 7   Conclusion

In this work, we present *DipNet*, a fully end-to-end policy for the strategy board game No Press Diplomacy. We collect a large dataset of human games to evaluate our architecture. We train our agent with both supervised learning and reinforcement learning self-play. Our tournament results suggest that DipNet is able to beat state-of-the-art rule-based bots in the No Press setting. Our ablation study and coalition analysis demonstrate that DipNet can effectively coordinate units and cooperate with other players. We propose Diplomacy as a new multi-agent benchmark for dynamic cooperation emergence in a rich environment. Probably the most interesting result to emerge from our analysis is the difference between the SL agent (trained on human data) and the RL agent (trained with self-play). Our coalition analysis suggests that the supervised agent was able to learn to coordinate support

Table 5: Coalition formation: Diplomacy agents comparison

|  |  | X-support-ratio | Eff-X-support-ratio |
|---|---|---|---|
| Human Game | No Press | 14.7% | 7.7% |
|  | Public Press | 11.8% | 12.1% |
|  | Press | 14.4% | 23.6% |
| Agents Games | RL DipNet | 9.1% | 5.3% |
|  | SL DipNet | 7.4% | 10.2% |
|  | Board State Only | 7.3% | 7.5% |
|  | Without FiLM | 6.7% | 7.9% |
|  | Masked Decoder (No Board) | 12.1% | 0.62% |

orders while this behaviour appears to deteriorate during self-play training. We believe that the most exciting path for future research for Diplomacy playing agents is in the exploration of methods such as LOLA [36] that are better able to discover collaborative strategies among self-interested agents.

**Acknowledgments**

We would like to thank Kestas Kuliukas, T. Nguyen (Zultar), Joshua M., Timothy Jones, and the webdiplomacy team for their help on the dataset and on model evaluation. We would also like to thank Florian Strub, Claire Lasserre, and Nissan Pow for helpful discussions. Moreover, we would like to thank Mario Huys and Manus Hand for developing DPjudge, that was used to develop our game engine. Finally, we would like to thank John Newbury and Jason van Hal for helpful discussions on DAIDE, Compute Canada for providing the computing resources to run the experiments, and Samsung for providing access to the DGX-1 to run our experiments.

## Footnotes

[5]Researchers can request access to the dataset by contacting webdipmod@gmail.com. An executive summary describing the research purpose and execution of a confidentiality agreement are required.

[6]https://trueskill.org/

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

## A  Tournament and TrueSkill Score

To compute TrueSkill, we ran a tournament where we randomly sampled a model for each power. For each game, we computed the ranks by elimination order (first power eliminated is 7th, second eliminated is 6th, ...), and the surviving powers by number of supply centers. We computed our Trueskill ratings using 1,378 games. We used the available python package[6], and the default TrueSkill environment configuration. The initial TrueSkill $\sigma$ is set to 8.33, and after 1,378 games the $\sigma$ is 0.64, which shows that the scores have converged.

Note that in our current evaluation settings we do not consider the existing power imbalance in the game, e.g., winning as Austria is harder than winning as France. Using a more sophisticated evaluation which includes the prior on the role of players is an interesting topic for future work.

## B  Effects of Graph Convolution Layers

We tested the affect of graph convolution layers by varying the number of layers in Table 6. After 8 layers of GCN there is no further improvement. We think this could be related to the fact that in the standard map of Diplomacy the most distant locations are connected by paths of length 8.

Table 6: Effect of GCN Layers

| Model | Accuracy per unit-order | | Accuracy for all orders | |
| --- | --- | --- | --- | --- |
| | Teacher forcing | Greedy | Teacher forcing | Greedy |
| DipNet | 61.3% | 47.5% | 23.5% | 23.5% |
| 8 GCN Layers | 61.2% | 47.4% | 23.4% | 23.4% |
| 4 GCN Layers | 61.1% | 47.2% | 23.2% | 23.2% |
| 2 GCN Layers | 60.3% | 45.9% | 23.2% | 23.2% |

## C  Effects of Decoding Granularity

We experimented with different decoding granularity. Instead of decoding each unit order as an atomic option (e.g. 'A PAR H'), we can decode as a sequence (e.g. ['A', 'PAR', 'H']). We call the first one unit-based and the latter token-based. We find that although the token-based model had better performance in terms of token accuracy, it had lower unit accuracy. We also try the transformer-based decoder. The results are in Table 7

Table 7: Comparison of different decoding granularity

| Model | Accuracy per unit-order | | Accuracy for all orders | | Accuracy per token | |
| --- | --- | --- | --- | --- | --- | --- |
| | Teacher forcing | Greedy | Teacher forcing | Greedy | Teacher forcing | Greedy |
| LSTM (order-based) | **61.3%** | **47.5%** | **23.5%** | **23.5%** | 82.1% | **74.4%** |
| Transformer (order-based) | 60.7% | **47.5%** | 23.4% | 23.4% | 81.9% | **74.4%** |
| LSTM (token-based) | 60.3% | 46.7% | 23.2% | 23.2% | **90.6%** | 73.8% |
| Transformer (token-based) | 58.4% | 45.4% | 22.3% | 22.3% | 90.0% | 72.9% |

