[Reviews · NeurIPS 2019]

Reviewer 1



originality: Moderate, the task have being studied in previous negotiation competition like D-brane, but the SL+ RL approach to study this problem is new. quality: Experiments are very solid, results are well presented. However, more theoretical analysis of the problem and intellectual insight would hugely improve the work. clarity: Well organized writing. However, the paper definitely does not provide enough information for possibility to reproduce the results. significance: Moderate. The approach that by collecting a large dataset of human games and do SL on it, later improve with RL is not new, but the execution of this idea on diplomacy is still a non-trivial job. This paper looks like a well-written, well-executed project report to me, though it should give either more insights, theories or more detailed algorithm, dataset and code.

Reviewer 2



The dynamically changing alliances mean that the domain of diplomacy presents unique challenges for agents. I agree with the authors that this means that diplomacy is ‘deserving of special attention’, I would consider the full game to be a grand challenge for multi-agent research. With recent progress in large-scale RL focusing on single-agent and 2-player zero sum games, this problem is particularly timely. This work presents state of the art agents trained with deep learning. To my knowledge this is the first successful application of deep learning to diplomacy. The design of the neural network is given particular attention: Several of the design choices are studied by ablation in the supervised task. The dataset of games is large, I’m not sure how many online games of diplomacy have been played, but 150,000 seems likely to be a significant proportion of them. Not all the games are No-Press, so there is a domain mis-match for much of the dataset. Maybe performance could be improved by including press rules as a feature for the network, or training more on the no-press games. I would like to see more details on how the dataset was selected. Sometimes in online games players fail to enter orders, or cheat by creating multiple accounts. Have any preprocessing steps been taken to remove data where these things have occurred? To evaluate the performance of agents, the SL DipNet played games against 4 rule-based baselines, including previous SOTA, plus the RL DipNet, under the conditions of 1 SL Dipnet vs 6 of the opponent, and 1 of the opponent vs 6 SL DipNet. Trueskill is then calculated based on these games. These 1 vs 6 matches are interesting to see, and clearly demonstrate that the DipNet programs outperform the other agents. However, we _cannot_ draw conclusions about the relative strength of the RL and SL agents from this tournament. The RL agent has had access to the SL agent to train against by the warm-start procedure, and the experiment never compares the RL DipNet against the baselines. I’d like the Trueskill to be generated by some more varied structure of games: A simple procedure that would be interesting is where each game, the player to play each power is sampled iid uniformly from the 6 different agents. Then we could understand how the agents perform against varied and heterogeneous opponents. This would test an important question in diplomacy, of whether self-play RL in 3+ agent problems overfits to its own play-style. The coalition analysis is nice, and I am particularly interested in the analysis around the effect of RL training on effective cross power support. One point to note is that cross power support in human no-press is likely to be reduced by virtue of signalling actions, where a move that is guaranteed to succeed is nonetheless supported by another player to signal that they want to ally, and help with coordination. I wonder if a metric for this behaviour can be designed, whether the SL network is able to pick up on such signalling moves, and whether RL changes this ability? Minor Comments: Line 28: Players can have a maximum of 17 units, because if they have 18+ units, they have already won the game. How was the average of 26 possible actions per unit calculated? Is that the empirical average in a dataset after removing any support/convoy orders that use non-existent units? Line 135: Did this mean to refer to [36] Learning with Opponent Learning Awareness, rather than the DIAL paper? Line 175: “there is no weight sharing”: Does this mean that there is no sharing of weights for the convolutions at different nodes of the graph, or just none between different layers? Would be good to clarify. ======= Comments following Author Response: Thanks to the authors for their clarification on how Trueskill was calculated. This makes it much easier to understand what the scores mean. There's some additional information I'd like to see included in the final paper if possible: - Table of results - Gameplay data (i.e. example games) - Report the standard deviations of Trueskill estimates

Reviewer 3



Comments after rebuttal and discussion ====================================== Thank you for your rebuttal. It appears you have foreseen many of the concerns and there simply isn't a great solution for many of them. It may be helpful to note in the body of the paper that the particular evaluation suffices here because the skill gap is so large in the agent pool. In the future as the pool of agents gets stronger a different evaluation may be necessary. Original Review =============== This paper is difficult to evaluate as a whole. I lean towards acceptance, but primarily because this paper may serve as a catalyst for future work. DipNet and its evaluation are not as thorough as one would hope for. The game of Diplomacy is well described and motivated as a challenging multi-agent AI domain. It is clear that it possess interesting aspects that are not represented in popular games in the literature---specifically the competitive/cooperative aspect as well as its communication structure. This paper is not the first to consider diplomacy as the authors point out, but past work, like Shapiro et al, was perhaps too early to spur academic interest. DipNet itself is non-trivial, but it is not incredibly insightful on its own. e.g., the state representation is domain-specific and the LSTM action decoding is heuristic. Bootstrapping training from human data and tuning with self-play is common practice. i.e., there are many seemingly important details, none of which are novel. The quality of this paper's contribution is mixed. On one hand, the motivation and exposition of Diplomacy as a test domain are excellent. The development and evaluation of DipNet is lacking, though. As mentioned above, the DipNet agent is fairly complicated, i.e., numerous non-trivial decisions were involved in its creation, many of which are parametrized. Typically, these decisions are not thoroughly justified and it is not clear how reliant the agent's performance is on these decisions. e.g., the state representation is lossy. The authors state "Based on our empirical findings, orders from the last movement phase are enough to infer the current relationship between the powers", but no evidence or procedure to reproduce this claim is provided. Similarly, using the no-check version of the game is claimed to be important to enable communication. This seems intuitive, but no evidence is provided to show this is indeed true or to validate the magnitude of this. Other decisions, such as the network structure, LSTM action decoding order, training procedure and parameters, reward and reward shaping, require further details and justification to aid in reproducibility. Perhaps the most important criticism of the paper is its evaluation of DipNet. The 1 vs. 6 head-to-head play demonstrates that DipNet is much stronger than the baselines, but the same approach is likely to fail when evaluating agents that are closer in strength. In multi-agent scenarios, is often the case that an agent trained with self-play will learn to implicitly collude with itself. e.g., if an agent indeed finds a Nash equilibrium then the solo agent will be at a tremendous disadvantage. This can even occur with different agents that trained in a similar fashion, e.g., playing 3 copies of agent A vs. 2 copies of agent B vs. 2 copies of agent C may not be informative if A and B are similar. i.e., using overall utility or games won as a performance indicator is incredibly sensitive to the pool of agents, especially in multi-agent scenarios. A thorough and thoughtful evaluation is of particular importance here as it sets a precedent for future papers. Again, I appreciate that the authors are willing to release their source code as it enables others to perform a different evaluation in the future should a more suitable one come to light. Minor comments: 269: metod => method Please go through the citations to correct capitalization, e.g., [15] nature => Nature

[Author Response · NeurIPS 2019]

We want to thank the reviewers for their helpful comments. The main purpose of this work is to lay the foundation for future research in Diplomacy, starting with the no-press setting. We regret that the page limit necessarily meant that we could not provide all the empirical details in the paper itself. In this response, we provide clarification regarding the **dataset release**, **tournament evaluation**, **architectural design**, **input representation**, and **other insights**.

**Dataset Release:** The dataset will be made available to any interested researchers. We are unable to make it available as a simple downloadable file due to privacy concerns (the data comes from users of an online Diplomacy service). We are working with the data owners on anonymizing the data to enable wider release. We will edit the sentence in the paper referring to the dataset availability to say "Researchers can access to the dataset by contacting user@domain.com."

**Tournament Evaluation:** In this work, we conducted **two forms of tournament settings**. In the first setting, we sample two models A and B, use A as one randomly chosen power and B plays the other six powers (1v6). In the second setting, we uniformly sample all models per game per power (TrueSkill). The bulk of Table 4 is from the first setting while our TrueSkill scores were computed with the second setting. We apologize for the confusion and can clarify this in the main text and add details regarding TrueSkill in an appendix. Importantly, **the TrueSkill scores support our claim that both RL DipNet (27.4) and SL DipNet (28.1) can beat the strongest rule-based approach, Albert (24.5)**, which is consistent with the one-vs-six results. In terms of whether RL DipNet or SL DipNet is better, we considered both settings (1v6 and TrueSkill). The two models have TrueSkill score within one standard deviation (0.62), but the one-vs-six result suggests RL DipNet is superior. We agree with R3's concerns about whether 1v6 is a comprehensive evaluation and R2's comment that it is difficult to draw meaningful conclusions regarding which is better. We considered a range of metrics, but many, including ELO and Glicko are designed for 2-player games and generalizing them to a 7-player game like Diplomacy is non-trivial. 1v6 has the benefit that it is more efficient to compute than TrueSkill, while TrueSkill is a well-studied and robust off-the-shelf evaluation metric. We believe that developing an efficient and accurate multi-agent evaluation is an open research question and Diplomacy offers an exciting testbed for this direction.

**Architectural Design:** We agree with R3 that there are a lot of non-trivial modeling choices in our architecture. We showed the effect of major design choices in our ablation study in Tables 2 and 3, e.g. **the effect of without order history**, **the effect of feeding an averaging embedding instead of a location-specific embedding to the decoder**, etc. We have also conducted more fine-grained experiments. We tried **different numbers of graph convolution layers (GCN)**, and we found that after 8 layers of GCN there is no further improvement. We think this could be related to the fact that in the standard map of Diplomacy the most distant locations are connected by a paths of length 8. We also tried using **order history with more than one year**, but it did not give much improvement in terms of accuracy. We also tried **different sequential orders** during decoding, and found that using a sequential order that does not jump across the map prevents performance dropping when decoding a long sequence with multiple units. We hypothesize that this is because a unit's orders are more influenced by nearby units than distant ones. We also tried **different decoding granularity**. Instead of decoding each unit order as an atomic option (e.g. 'A PAR H'), we can decode as a sequence (e.g. ['A', 'PAR', 'H']). We call the first one unit-based and the latter token-based. We found that although the token-based model had better performance in terms of token accuracy, it produced fewer correct unit orders overall. We apologize for writing some of the claims without referring to the evidence, like "orders from the last movement phase are enough to infer the current relationship between the powers". We will properly modify it and provide more results in appendix.

**Input Representation** Our input representation is a result of both empirical findings and domain knowledge. Because we want to take advantage of the game adjacency map, we treat each location on the map as a node and try to encode the unit information as a node feature. We also have special treatment for the coast. E.g., if a fleet is on a coast, we will also encode that fleet's information on the parent location, because in Diplomacy occupying any coast of the location is equivalent to occupying the location. We also tried **different ways of encoding history information**. Previously we encoded the history by appending the board state from the previous year, but that did not help much. We hypothesize that the order history should be more informative than board state history because order history captures additional information (e.g., attacks that failed, supports and convoys.). This information can convey relationships between powers. Given this, we encode history by embedding the previous orders on the map as an extra node feature.

**Other** In terms of RL training, our main insight is that reward shaping based on gaining and losing supply centers is helpful in training. Both R2 and R3 also raise the question of whether there is an effective way to detect **signaling support orders**, particularly in no-check games. Although we have not done such an analysis yet, one simple method to see how common they are would be to compare the fraction of orders that are invalid in no-press and press games. Since signaling orders with invalid syntax are predominantly used in no-press games, we should see a large gap in the measurements. However a case-by-case analysis is still needed to differentiate signalling from mere mistakes.

[Meta-Review · NeurIPS 2019]

All reviewers agree that this paper explores interesting territory, i.e., multi-agent Learning in the Diplomacy game. It is a well written and presented paper. The paper has generated quite some discussion after the rebuttal, discussing all pros and cons of the work. The major point in favor of the work (as also indicated by the authors themselves) seems to be that the work lays some ground work for future research in the Diplomacy game, that is known to be very hard and challenging. The biggest point of concern is that the paper presents little innovation in the techniques that it deploys but rather shows how the SOTA can be used/engineered to be successful in this domain to a certain extent, and illustrates the performance of known algorithms. The importance of this work is that it lays some ground work for future research to build on this initial study. There is consensus that this is indeed important. There are still many unanswered questions though about the performance of DipNet, which require a lot more work to be carried out. The authors have not responded to the question of code release raised by reviewers, and that remains also a point of concern as well. As a follow-up on the evaluation issues the authors discuss (feedback/paper). Some of the reviewers felt it would be nice if they would add a small discussion on the topic as it is gaining quite some interest, see e.g. Balduzzi et al., and Omidhshafiei et al. Note that the latter propose a method for evaluation that does apply to n-player games. - David Balduzzi et al. : Re-evaluating evaluation. NeurIPS 2018: 3272-3283 - Shayegan Omidshafiei et al.: α-Rank: Multi-Agent Evaluation by Evolution. Scientific Reports, 2019